# Ginsenoside Rg3 Prevents Oncogenic Long Noncoding RNA ATXN8OS from Inhibiting Tumor-Suppressive microRNA-424-5p in Breast Cancer Cells

**DOI:** 10.3390/biom11010118

**Published:** 2021-01-18

**Authors:** Heejoo Kim, Hwee Won Ji, Hyeon Woo Kim, Sung Hwan Yun, Jae Eun Park, Sun Jung Kim

**Affiliations:** Department of Life Science, Dongguk University-Seoul, Goyang 10326, Korea; heejoo0923@dongguk.edu (H.K.); hweewon96@dongguk.edu (H.W.J.); opopr5@dongguk.edu (H.W.K.); skskbby@dongguk.edu (S.H.Y.); 201511717@dongguk.edu (J.E.P.)

**Keywords:** ceRNA, CpG methylation, ginsenoside Rg3, long noncoding RNA, microRNA

## Abstract

Ginsenoside Rg3 exerts antiproliferation activity on cancer cells by regulating diverse noncoding RNAs. However, little is known about the role of long noncoding RNAs (lncRNAs) or their relationship with competitive endogenous RNA (ceRNA) in Rg3-treated cancer cells. Here, a lncRNA (ATXN8OS) was found to be downregulated via Rg3-mediated promoter hypermethylation in MCF-7 breast cancer cells. SiRNA-induced downregulation of ATXN8OS decreased cell proliferation but increased apoptosis, suggesting that the noncoding RNA possessed proproliferation activity. An in silico search for potential ATXN8OS-targeting microRNAs (miRs) identified a promising candidate (miR-424-5p) based on its high binding score. As expected, miR-424-5p suppressed proliferation and stimulated apoptosis of the MCF-7 cells. The in silico miR-target-gene prediction identified 200 potential target genes of miR-424-5p, which were subsequently narrowed down to seven that underwent hypermethylation at their promoter by Rg3. Among them, three genes (EYA1, DACH1, and CHRM3) were previously known oncogenes and were proven to be oppositely regulated by ATXN8OS and miR-424-5p. When taken together, Rg3 downregulated ATXN8OS that inhibited the tumor-suppressive miR-424-5p, leading to the downregulation of the oncogenic target genes.

## 1. Introduction

Ginsenoside Rg3 is a steroidal saponin derivative that is abundant in heat-processed ginseng extract [1]. Rg3 possesses potent anticancer properties and is known to modulate diverse cellular events such as cell proliferation, immune response, autophagy, metastasis, and angiogenesis [2]. Rg3 activates proapoptotic proteins such as caspase-3 and Bax but suppresses antiapoptotic protein Bcl-2 [3]. In the process, NF-κB, which drives cell-cycle progression, is inhibited by blocking the phosphorylation of Akt and ERK kinases [4]. In MDA-MB-231 breast-cancer cells, Bcl-2 can be suppressed by destabilizing a mutant P53 with Rg3 [4]. In osteosarcoma cell lines, Rg3 inhibits migration and invasion by suppressing MMPs and the Wnt/β-catenin pathway, which are related to epithelial-mesenchymal transition (EMT) and angiogenesis [5]. Rg3-treated gastric cancer cells show a remarkably lower expression of HIF-1α and VEGF under hypoxia [6]. The SNAIL signaling axis is another key pathway regulated by Rg3 during metastasis, which regulates EGFR and fibronectin in cancer stem cells [7].

Rg3 can inhibit cancer-cell growth by modulating epigenetic factors of oncogenes or tumor suppressors. A genome-wide methylation analysis identified over 250 genes with significant changes in methylation level at specific CpG sites in Rg3-treated MCF-7 breast-cancer cells [8]. These genes were largely associated with cell-morphology-related pathways. Notably, NOX4 and KDM5A were hyper- and hypo-methylated on their promoter regions, respectively, which led to gene dysregulation and increases in cell apoptosis [8]. Other genes such as p53, Bcl-2, and EGF were affected by Rg3-mediated promoter methylation in the HepG2-hepatocarcinoma cell line [9]. Approximately a dozen (microRNAs) miRs are known to be regulated by Rg3, many of which are involved in cancer malignancy, metastasis, or EMT [10,11]. For example, miR-145 comprises the DNMT3A-miR-145-FSCN1 axis in ovarian cancer, and its downregulation by Rg3 inhibits EMT [12]. Recently, miRs associated with the Warburg effect [13] and autophagy [14] were identified as Rg3 targets. Rg3 upregulated miR-519a-5p via reducing DNMT3A-mediated DNA methylation to inhibit an HIF-1α-stimulated Warburg effect in ovarian cancer [13]. MiR-181b impaired the antiautophagy effect of Rg3-mediated tumor cytotoxicity by modulating the CREBRF/CREB3 signaling pathways in gallbladder cancer [14].

LncRNAs (i.e., noncoding RNAs larger than 200 nucleotides) are known to regulate a variety of genes, leading to tumor-development stimulation or suppression [15]; however, only a few lncRNAs have been identified as Rg3 targets. LncRNA-CASC2 is upregulated by Rg3, thereby activating PTEN signaling and suppressing drug-resistant pancreatic cancer cells [16]. Two tumor-related lncRNAs (RFX3-AS1 and STXBP5-AS1) have been identified in Rg3-treated MCF-7 cells, and their expression is controlled by promoter methylation [17]. Moreover, lncRNA CCAT1 induces Caco-2 colorectal-cancer-cell proliferation but is also downregulated by Rg3 [18].

A number of epigenetic factors have been found to act in conjunction to regulate the expression of specific target genes. Moreover, competitive endogenous RNA (ceRNA) sponges miR by sharing the same target gene recognition sequence [19]. For example, lncRNA H19 acts as a miR-340-3p sponge to promote epithelial-mesenchymal transition in breast-cancer cells [20], thereby disrupting the gene-suppression activity of miR. Although ginsenosides are known to regulate miRs and lncRNAs in cancer cells, few studies have characterized the role of ceRNA. In this study, a genome-wide methylation-array dataset was analyzed to identify lncRNAs that were epigenetically regulated by Rg3. Notably, the lncRNA ATXN8OS was found to be hypermethylated by Rg3 in MCF-7 breast-cancer cells. The effect of Rg3 on ATXN8OS expression was then examined, and the role of the lncRNA in cancer-cell growth was elucidated. A miR that interacts with ATXN8OS was examined to identify sponge-activity relationships between the two RNAs during miR-mediated gene regulation in the presence of Rg3.

## 2. Materials and Methods

### 2.1. Cell Culture

Human mammary-gland-derived cell lines (MCF-10A, MCF-7, and MDA-MB-231) were purchased from the American Type Culture Collection (ATCC, Manassas, VA, USA). MCF-10A was cultured in MEGM (Lonza, Basel, Switzerland) with 100 ng/mL cholera toxin. MCF-7 and MDA-MB-231 were cultured in RPMI 1640 medium (Welgene, Seoul, Korea) supplemented with 10% fetal bovine serum (Capricorn Scientific, Ebsdorfergrund, Germany). All cells were supplemented with 2% penicillin/streptomycin (Capricorn Scientific) and cultured at 37 °C with 5% CO_2_ in a humidified incubator.

### 2.2. Rg3 Treatment and Transfection

5 × 10^4^ cells were seeded in a 60 mm culture dish with 50% confluence and cultured for 24 h before Rg3 treatment or transfection. The cells were then treated with 20 and 50 µM of Rg3 using a 20 mM Rg3 stock (LKT Labs, St. Paul, MN, USA) in 100% ethanol. For transfection, siRNA (Bioneer, Daejon, Korea), mimic miR (Bioneer), and inhibitor miR (Bioneer) were diluted to final concentrations of 20 and 40 nM in Opti-MEM Medium (Invitrogen, Carlsbad, CA, USA), mixed with 5 μL of Lipofectamine RNAiMAX (Invitrogen), and added to the cell culture. For Rg3 and RNA cotreatments, RNA was processed following the aforementioned transfection protocol, and, after 24 h, Rg3 was added. The cells were further cultured for 24 h and then harvested using 0.05% trypsin-EDTA (Gibco BRL, Carlsbad, CA, USA).

### 2.3. Rg3-Quantitative Reverse-Transcription Polymerase Chain Reaction (qRT-PCR)

Chromosomal DNA and total RNA were extracted from the 60 mm culture dishes using the ZR-Duet DNA/RNA MiniPrep kit (Zymo Research, Irvine, CA, USA) and eluted to 50 and 20 μL, respectively. MiR cDNA was synthesized from 1 μg of total RNA using a miScript II RT kit (Qiagen, Valencia, CA, USA) in 20 μL reactions. qRT-PCR was conducted with 3 μL cDNA per reaction using the miScript SYBR Green PCR kit (Qiagen) and miScript Primer Assay kit (Qiagen). mRNA cDNA was synthesized from 2 μg of total RNA using ReverTra Ace qPCR RT Master Mix (Toyobo, Osaka, Japan) in 10 μL reactions. PCR was then conducted from 1 μL cDNA using SYBR Fast qPCR Kit Master Mix (Kapa Biosystems, Wilmington, MA, USA). The expression of miR and mRNA samples was normalized to that of U6 and glyceraldehyde-3-phosphate dehydrogenase (GAPDH), respectively. PCR was performed with an ABI 7300 instrument (Applied Biosystems, Foster City, CA, USA), and the expression level was calculated following the 2^−ΔΔCt^ method. Methylation-specific PCR was performed with bisulfite-treated DNA, and the methylation level was calculated by the 1/[1+2^− (CTu−CTme)^] × 100% method, as previously described [21]. PCR primers are listed in Appendix A.

### 2.4. Data Mining

LncRNAs showing a significant methylation change by Rg3 were retrieved after analyzing the methylation-array data of the NCBI GEO DataSet (GSE99505). LncBase Predicted v.2 (http://diana.imis.athena-innovation.gr/DianaTools) and StarBase v3.0 (http://starbase.sysu.edu.cn/index.php) were used to identify miRs that potentially interact with ATXN8OS. MiR target genes were selected using five miR target-prediction programs: MicroT (www.microrna.gr/microT-v4), RNA22 (https://cm.jefferson.edu/rna22), TargetScan7 (http://www.targetscan.org/vert_72), miRWalk (http://http://mirwalk.umm.uni-heidelberg.de), and miRmap (https://mirmap.ezlab.org).

### 2.5. Cell Proliferation and Apoptosis Assay

The effect of Rg3 and noncoding RNAs on cell growth was analyzed by a dye-based cell-proliferation assay as previously described [22]. Briefly, 2 × 10^3^ cells were seeded per well on a 96-well plate and cultured for 24 h. Afterward, the cells were treated with either Rg3 or noncoding RNA and cultured for up to six additional days. After an appropriate culture period, the cells were stained with WST-8 using the Cell Counting Kit-8 (CCK-8) (Enzo Biochem, New York, NY, USA) to measure cell density at OD_450_ using a spectrophotometer. For the apoptosis analysis, 1 × 10^6^ cells were seeded in a 60 mm plate, treated with Rg3 or transiently transfected with siRNA, and cultured for 24 h. After harvesting, 1 × 10^5^ cells were suspended in a 1x binding buffer provided with the Annexin V-FITC Apoptosis Detection kit II (BD Bioscience, San Jose, CA, USA), then stained with FITC Annexin V(BD Bioscience) and PI (Sigma-Aldrich, St. Louis, MO, USA). Fluorescence was detected with a BD Accuri C6 flow cytometer (BD Bioscience), and the data were analyzed with the BD Accuri C6 software (BD Bioscience). Cell-cycle analysis was performed using a flow cytometer as previously described [23]. The cell-proliferation index was calculated using the following formula: proliferation index = (S+G2+M)/(G0/G1+S+G2+M) × 100 (%), where each letter represents the number of cells at each stage.

### 2.6. Western Blot Analysis

Proteins were extracted from the harvested cells using ice-cold RIPA lysis buffer (Thermo Fisher Scientific, Waltham, MA, USA) with a 1% protease-inhibitor cocktail (Thermo Fisher Scientific). The proteins (15 μg) were then subjected to SDS-PAGE, blotted on a PVDF membrane (Sigma-Aldrich), and treated with primary antibodies overnight at 4 °C. The blot was then incubated with HRP-conjugated antirabbit IgG antibodies (1:1000, GTX213110-01; GeneTex, Irvine, CA, USA) for 2 h. The signals were visualized with the ECL reagent (Abfrontier, Seoul, Korea), quantified using the Image Lab software (Bio-Rad, Hercules, CA, USA), and normalized with β-actin. The antibodies used were anti-CHRM3 (1:1000, GTX111637; GeneTex), anti-DACH1 (1:1000, A303-556A-M; Bethyl, Montgomery, TX, USA), and anti-β-actin (1:1000, bs-0061R; Bioss, Woburn, MA, USA).

### 2.7. Statistical Analyses

All experiments were independently conducted in triplicate, and the results were expressed as the mean ± SD. Statistical analyses were performed using the SPSS 23.0 software (SPSS, Chicago, IL, USA). T-tests, originally created by Two-tailed Student, were performed to analyze the qRT-PCR, Western blot, and apoptosis assay results. *p*-value < 0.05 was considered statistically significant.

## 3. Results

### 3.1. Rg3 Induces Hypermethylation and Downregulation of ATXN8OS

We previously performed a genome-wide methylation analysis of Rg3-treated MCF-7 breast-cancer cells [8]. In addition to 866,895 CpGs in protein-coding genes, the array covered 10,733 CpGs in noncoding RNAs. Six lncRNAs exhibited significant methylation changes in the promoter (i.e., |methylation level change (Δβ)| > 1.5 and |methylation fold change| > 1.4) (Figure 1A). Given that many lncRNAs have been linked to the development of various cancer types, our study focused on their regulatory mechanisms. ATXN8OS was selected for further study as it exhibited the highest methylation level change (Δβ = 0.189). Although little is known about its role in cancer development and progression, previous studies indicate that ATXN8OS has oncogenic properties and therefore stimulates cancer-cell growth [24].

The induction of hypermethylation at the ATXN8OS promoter by Rg3 was verified via methylation-specific PCR in MCF-7 cells treated with 20 and 50 μM of Rg3. This experiment resulted in a similar methylation change (methylation-fold change = 1.4 and Δβ = 1.5) to that of the array-based analysis. Moreover, according to the qRT-PCR analysis, ATXN8OS was downregulated by up to 76% in the Rg3-treated MCF-7 cells (Figure 1B). As Rg3 is known to share a structural similarity with estrogen [25], regulation of ATXN8OS may be affected by the estrogen-receptor (ER) status. To test this, the effect was examined in an ER-negative breast-cancer cell line, MDA-MB-231, and in an ER-positive normal cell line, MCF-10A. The result showed that expression of ATXN8OS was less affected in MDA-MB-231 than in the other two cell lines (Appendix A), possibly implying an ER dependence on Rg3 for ATXN8OS regulation.

To address how ATXN8OS contributes to cancer-cell growth, its downregulation was induced using two siRNAs (siATXN8OS#1 and #2) in MCF-7, which targeted different sites of ATXN8OS (Appendix A, Appendix A), after which cell proliferation and apoptosis were monitored. It was found that ATXN8OS siRNA suppressed cancer-cell growth by up to 18%, increased apoptosis by up to 5%, and decreased the cell-proliferation index from 36.7% to 21.5% (Figure 1C–F; Appendix A). These results suggest that ATXN8OS promotes proliferation by stimulating the MCF-7 cancer-cell growth while also suppressing apoptosis.

### 3.2. ATXN8OS Stimulates Cancer-Cell Proliferation via Sponging miR-424-5p

LncRNAs are known to often interact with and regulate miRs and act as ceRNA to modulate the expression of miR target genes. Therefore, our study sought to identify potential miRs for ATXN8OS. Three candidates were identified upon screening the LncBase and StarBase public databases, which offer potential partner miRs for lncRNAs (Figure 2A). MiR-424-5p was selected for further analysis as it showed the highest binding score. Rg3 treatment in MCF-7 cells induced the upregulation of the miR (Figure 2B). To see whether ATXN8OS could regulate miR-424-5p, the expression of the miR was quantified via qRT-PCR in MCF-7 cells treated with ATXN8OS-specific siRNA (siATXN8OS). Compared to the scrambled siRNAs, siATXN8OS significantly increased the expression of miR-424-5p (Figure 2C). The expression of ATXN8OS was then examined after deregulating miR-424-5p using a mimic or an inhibitor RNA (Appendix A). Interestingly, the miR-424-5p mimic RNA downregulated ATXN8OS, whereas the inhibitor upregulated the lncRNA (Figure 2D).

Afterward, the effect of miR-424-5p on MCF-7 cell proliferation and apoptosis in the presence of Rg3 was examined after deregulating miR-424-5p in combination with Rg3. As shown in Figure 3A, cell growth was suppressed by 30% using the miR mimic alone, and further decreased by Rg3 exposure in a dose-dependent manner. The miR mimic increased apoptosis by 15% (Figure 3B). In contrast, the miR-424-5p inhibitor reversed the effect of the mimic RNA by increasing cell growth while decreasing apoptosis of MCF-7 (Figure 3C,D). Therefore, we concluded that Rg3 inhibited the proproliferation effect of the miR-424-5p inhibitor.

### 3.3. MiR-424-5p Target Genes are Regulated by ATXN8OS

Given the regulatory effect of miRs on target genes, we sought to determine whether ATXN8OS also affects target-gene expression. Potential targets were first identified using the five target-gene prediction algorithms described in the Materials and Methods, which rendered 200 candidate genes according to all five prediction tools (Figure 4A). To narrow down the number of target genes, the pool was then further filtered by applying genome-wide methylation-array data, which were obtained from the Rg3-treated MCF-7 cells (GSE99505). We aimed to identify target genes that were controlled by miR-424-5p and subject to promoter methylation by Rg3. Through this double-filtering approach, seven genes were identified, satisfying both the target-gene prediction and the methylation criteria (|Δβ| > 1.5) (Figure 4B). Specifically, our study focused on EYA1, CHRM3, and DACH1 because they had a target sequence for miR-424-5p (Figure 4C) and showed hypermethylation in the array data, suggesting that they were downregulated by Rg3. Additionally, these three genes had previously been reported to possess oncogenic properties in several cancer types [26,27], except DACH1, which functioned as either a tumor promoter [28] or suppressor [29] depending on the cancer type. Consistent with the hypermethylation status, EYA1, CHRM3, and DACH1 were downregulated by 39–95% by Rg3, as determined by our qRT-PCR assays (Figure 4D). ATXN8OS inhibition resulted in downregulation of all the target genes (Figure 4E). Moreover, the miR-424-5p mimic downregulated the three target genes, whereas an inhibitor upregulated them (Figure 4F,G).

The protein expression of DACH1 and CHRM3 was then examined by Western blot analysis. DACH1 and CHRM3 protein-expression exhibited a similar profile to that of the transcripts. Specifically, protein expression was downregulated by Rg3, siATXN8OS, and a miR-424-5p mimic RNA but upregulated by the miR-424-5p inhibitor (Figure 5 and Appendix A). The EYA1 protein was barely detected in MCF-7 as in a previous study [30]. Therefore, further confirmation of the effect of Rg3 and noncoding RNAs at the protein level was deemed unnecessary. Overall, Rg3 downregulated EYA1, DACH1, and CHRM3 via the Rg3/ATXN8OS/miR-424-5p axis, whereas ATXN8OS inhibited the miR to modulate the expression of the target gene (Figure 6).

## 4. Discussion

Our study aimed to identify lncRNAs that are dysregulated in Rg3-treated cancer cells to elucidate the mechanisms by which they control cancer-cell proliferation, with a particular focus on ceRNA-miR interaction. Most studies on ATXN8OS have so far examined the genetic expansion of CAG repeats. For instance, spinocerebellar ataxia type 8 (SCA8), an autosomal dominant neurodegenerative disease, is caused by CTA/CTG repeat expansion in the ATXN8OS gene [31]. In contrast, little is known about the role of ATXN8OS in tumor development. Recently, Deng et al. found that ATXN8OS stimulated the proliferation and migration of MCF-7 and MDA-MB-231 breast-cancer cells [24]. Specifically, the authors reported that ATXN8OS sequestered the tumor-suppressive miR-204. However, the mechanisms by which miR-204 is regulated by Rg3 remain to be determined. Our study revealed that the oncogenic ATXN8OS is epigenetically regulated by Rg3 via promoter methylation. A few other lncRNAs also showed methylation level changes: DOCK4-AS1, LINC00911, and RFX3-AS1 were hypermethylated, whereas STXBP5-AS1 and LINC01477 were hypomethylated. Notably, LINC00911 and RFX3-AS1 are known as oncogenes [17,32], whereas STXBP5-AS1 is known as a tumor suppressor [33]. These findings suggest that the tumor-suppressive activity of Rg3 could be attributed in part to its epigenetic regulation of tumor-related lncRNAs. However, the mechanisms by which ATXN8OS methylation is controlled by Rg3 remain to be determined. Moreover, although a close association was identified between gene methylation and expression levels, additional studies are required to determine whether inducing hypermethylation could drive gene downregulation.

MiR-424-5p has been shown to reduce cell viability by modulating the PTEN/PI3K/AKT/mTOR pathway in breast-cancer cells [34], the MAPK pathway in ischemic stroke [35], and the Hippo-signaling pathway in thyroid cancer [36]. MiR-424-5p target genes have been identified in various cancer types, including PD-L1 [34], VEGFA [37], and ARK5 [38]. These target genes generally exert a protumor activity by promoting proliferation, migration, or angiogenesis in cancer cells. A few lncRNAs have been found to regulate miR-424-5p in various cancer cells, including LINC00922 in breast cancer [39], CDNK2B-AS1 in hepatocellular carcinoma [40], and XIST in neuroendocrine tumors [41]. In all the aforementioned cases, regulation of miR-424-5p by the corresponding lncRNA resulted in cell proliferation or cancer-progression alterations.

Limited cases of ceRNA have been identified in ginsenosides. However, there are reports of an Rg3-regulated lncRNA H19 that sponges miR-324-5p to enhance PKM2 expression by directly binding the miR [42]. In another study, Rg1 inhibited high glucose-induced mesenchymal activation by downregulating lncRNA RP11-982M15.8 but upregulating miR-2133 to decrease Zeb1 [43]. The current study suggests a novel ceRNA relationship between the Rg3-regulated ATXN8OS and miR-424-5p, which is supported by the following findings: First, the expression of miR-424-5p increased after ATXN8OS was inhibited and *vice versa.* The lncRNA-induced miR regulation may increase through binding sites with special sequences or paring topology, which would trigger miR degradation upon binding [44]. Second, the two noncoding RNAs had opposite effects on the target-gene expression and the MCF-7 cell growth. Nonetheless, the mechanical interaction between the two RNAs should be elucidated to confirm the proposed ceRNA relationship.

Our study had a few noteworthy limitations. Particularly, all of our findings were based on the analysis of a single lncRNA. Therefore, data on lncRNAs other than ATXN8OS should be obtained to comprehensively explore how Rg3-regulated lncRNAs affect cancer-cell survival or proliferation. Additionally, further studies on other lncRNAs identified herein such as RFX3-AS1, DOCK4-AS1, and STXBP5-AS1 could provide useful insights.

## 5. Conclusions

ATXN8OS was identified as a lncRNA that can be downregulated via promoter hypermethylation by Rg3 in MCF-7 cancer cells. Moreover, ATXN8OS was found to induce the proliferation of cancer cells and this was suppressed by Rg3. At the molecular level, ATXN8OS sponged a tumor-suppressive miR-424-5p, thereby activating key oncogenes such as EYA1, DACH1, and CHRM3, which could be suppressed by Rg3 treatment. Therefore, our findings suggest that Rg3 suppresses MCF-7 cancer-cell proliferation but increases apoptosis by modulating the ATXN8OS/miR-424-5p/target-gene axis.

## Figures and Tables

**Figure 1 biomolecules-11-00118-f001:**
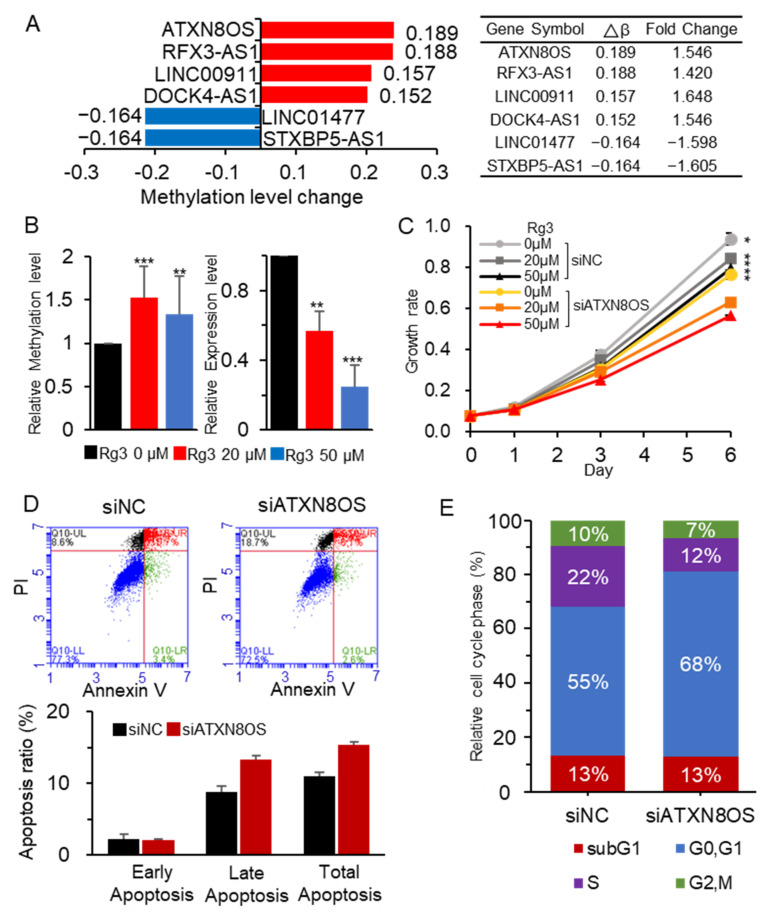
ATXN8OS with proproliferation activity in the MCF-7 cells was downregulated by Rg3 via promoter methylation. (**A**) ATXN8OS was among the six lncRNAs that exhibited significant changes in methylation level (|Δβ| ≥ 0.15 and |fold change| ≥ 1.4), as demonstrated by the analysis of an Rg3-treated MCF-7-cell methylation array. (**B**) MCF-7 cells were treated with 20 and 50 μM of Rg3, and the methylation and expression of ATXN8OS were examined by methylation-specific PCR and qRT-PCR, respectively. (**C**) ATXN8OS was downregulated in MCF-7 using siRNA, and its effect on cell proliferation was examined in the presence of Rg3 using the CCK-8 assay. (**D**,**E**) The effect of ATXN8OS on apoptosis (**D**) and cell cycle (**E**) was monitored using flow cytometry. All experiments were performed in triplicate, and the values are presented as the mean ± SE. siNC, control siRNA (40 μM); siATXN8OS, ATXN8OS-specific siRNA (40 μM). * *p* < 0.05, ** *p* < 0.01, *** *p* < 0.001.

**Figure 2 biomolecules-11-00118-f002:**
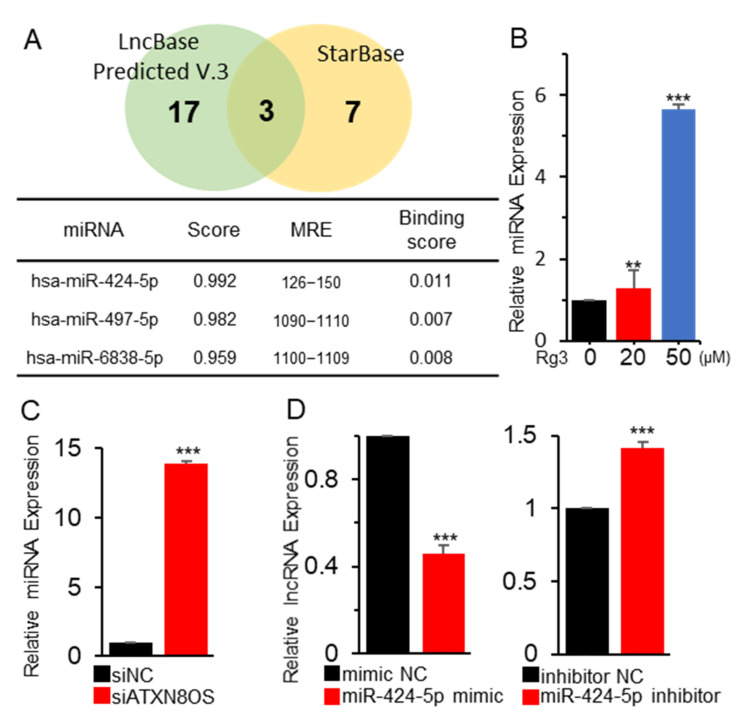
ATXN8OS and miR-424-5p sponge each other. (**A**) Three miRs that could potentially bind ATXN8OS were screened in silico using two miR-prediction databases (LncBase Predicted v.3 and StarBase). (**B**) miR-424-5p exhibited the highest binding score and was therefore examined to characterize its regulation by Rg3. MCF-7 cells were treated with Rg3, and the RNA expression was quantified by qRT-PCR. (**C**,**D**) The association between the ATXN8OS and miR-424-5p expression was monitored by examining the expression of each RNA after inhibiting ATXN8OS using siRNA (**C**) and overexpressing (40 μM) or inhibiting miR-424-5p (20 μM) (**D**). All experiments were performed in triplicate, and the values are presented as the mean ± SE. Testing was done using siNC, negative control siRNA (40 μM); siATXN8OS, ATXN8OS-specific siRNA (40 μM); mimic NC, negative control mimic for miR-424-5p (40 μM); and inhibitor NC, negative control inhibitor for miR-424-5p (20 μM). ** *p* < 0.01, *** *p* < 0.001.

**Figure 3 biomolecules-11-00118-f003:**
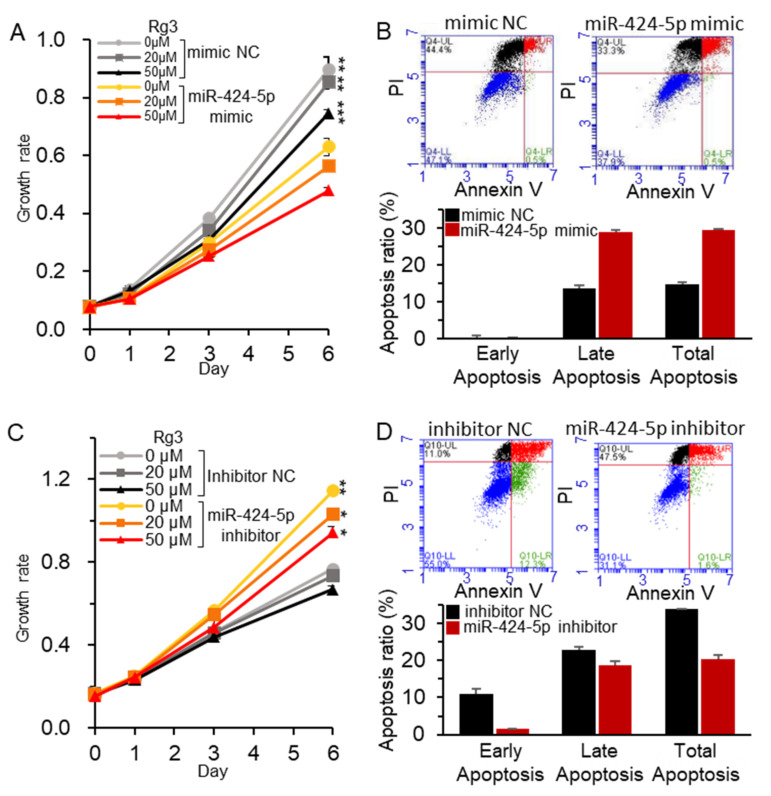
MiR-424-5p inhibited MCF-7 cell proliferation. MiR-424-5p was deregulated in MCF-7 by transiently transfecting the cells with a mimic (**A**,**B**) or an inhibitor (**C**,**D**), after which cell proliferation and apoptosis were assessed with the CCK-8 assay and flow-cytometry analysis. Rg3 was coadministered with the mimic (40 μM) or inhibitor (20 μM) for the proliferation assay. Testing was done using mimic NC, negative control miR-424-5p mimic (40 μM) and inhibitor NC, negative control inhibitor for miR-424-5p (20 μM). All experiments were performed in triplicate, and the results are presented as the mean ± SE. Representative images are shown for flow-cytometry analysis. * *p* < 0.05, ** *p* < 0.01, *** *p* < 0.001.

**Figure 4 biomolecules-11-00118-f004:**
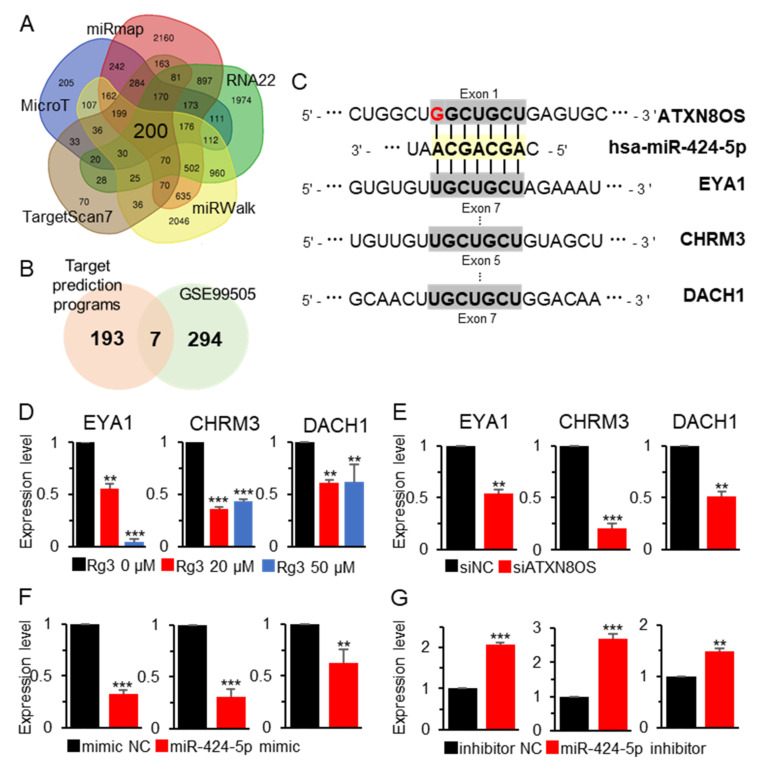
Regulation of miR-424-5p target genes by Rg3 and ATXN8OS. Potential miR-424-5p target genes were identified by analyzing five public databases (miRmap, miRWalk, TargetScan, MicroT, and RNA22) (**A**), after which they were compared with the methylation-array data of the Rg3-treated MCF-7 cells (GSE99505) (**B**). (**C**) Potential binding sequence of the target genes on miR-424-5p. The seed sequence is denoted in bold. (**D**–**G**) Effect of Rg3, ATXN8OS, and miR-424-5p on miR-424-5p target-gene expression. Gene expression was examined by qRT-PCR for samples treated with Rg3 (**D**), ATXN8OS-specific siRNA (40 μM) (**E**), miR-424-5p mimic (40 μM) (**F**), and a miR-424-5p inhibitor (20 μM) (**G**). Testing was done using siNC, control siRNA (40 μM); mimic NC, negative control mimic for miR-424-5p (40 μM); and inhibitor NC, negative control inhibitor for miR-424-5p (20 μM). All experiments were performed in triplicate, and the results are presented as the mean ± SE. ** *p* < 0.01, *** *p* < 0.001.

**Figure 5 biomolecules-11-00118-f005:**
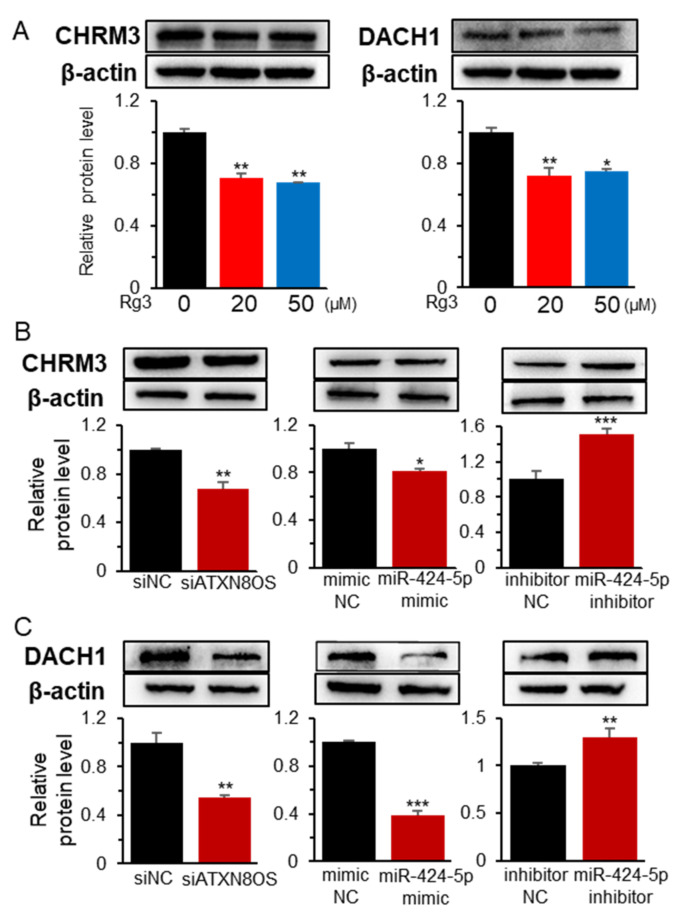
Effect of Rg3, ATXN8OS, and miR-424-5p on the target genes of miR-424-5p at the protein level. Western blot analysis of CHRM3 and DACH1 was performed after treating the MCF-7 cells with Rg3 (**A**) or deregulating ATXN8OS (40 μM siRNA) and miR-424-5p (40 μM for mimic and inhibitor) (**B**,**C**). Testing was done using siNC, control siRNA (40 μM); mimic NC, negative control mimic for miR-424-5p (40 μM); and inhibitor NC, negative control inhibitor for miR-424-5p (20 μM). The band intensity was measured with the Image Lab software and indicated by bar graphs. * *p* < 0.05, ** *p* < 0.01, *** *p* < 0.001.

**Figure 6 biomolecules-11-00118-f006:**
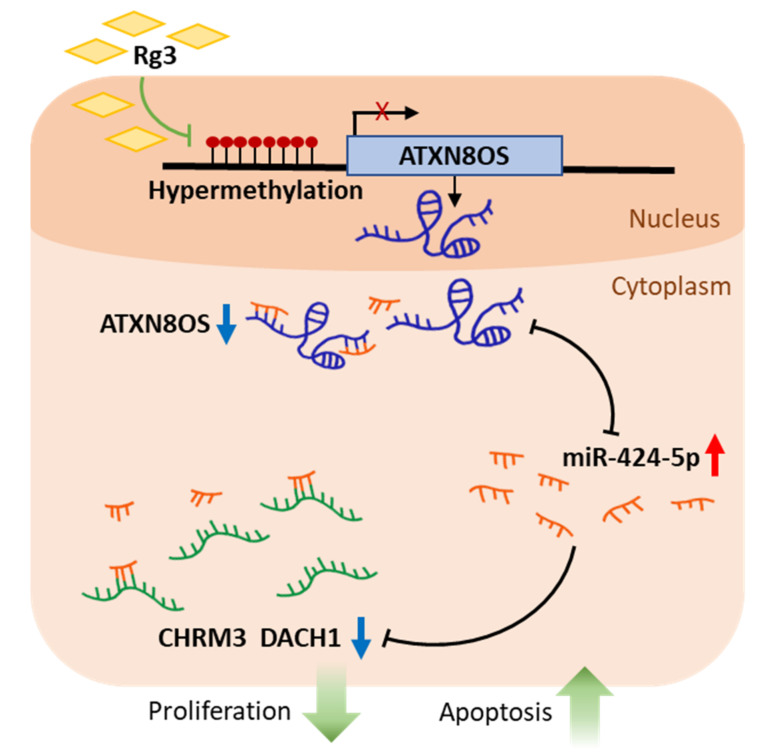
Schematic of the Rg3/ATXN8OS/miR-424-5p axis regulation process. ATXN8OS downregulates the tumor-suppressive miR-424-5p, which in turn activates oncogenic CHRM3 and DACH1, leading to cancer-cell proliferation. Rg3 blocks the oncogenic activity of ATXN8OS by inducing promoter hypermethylation.

## Data Availability

All data are contained within the article or supplementary material.

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
