# Peer review of "Ginsenoside Rg3 Prevents Oncogenic Long Noncoding RNA ATXN8OS from Inhibiting Tumor-Suppressive microRNA-424-5p in Breast Cancer Cells"

_biomolecules, 2021, doi:10.3390/biom11010118_

Round 1

Reviewer 1 Report

Authors have addressed all the comments well after few rounds of review. 

Reviewer 2 Report

Based on the data shown in this second revision, my concerns were addressed and solved

Reviewer 3 Report

The authors adressed to all the comments raised by the reviewers appropriately.  Thus, the manuscrip is acceptable for publication to Biomolecules.

This manuscript is a resubmission of an earlier submission. The following is a list of the peer review reports and author responses from that submission.

Round 1

Reviewer 1 Report

Line 54-55: Need to give some explanation as well. After identifying the targets what was the implication. 

The results and discussions are well written.

However, would need the clarification for certain points:

  1. Is there a reason in vivo study is not done?
  2. How about a control cell line. A normal cell line should be used as a control.
  3. What is the impact of the study on other cancers such as MDA-MB-231 and 4T1. 
  4. What happens if the concentration of Rg3 is less than 20 and more than 50 μM
  5. Western Blot: Is the relative density of the bands calculated with respect to housekeeping gene?

Author Response

Please refer to the attached file for the authors' response to the reviewer's comments.

Reviewer 2 Report

In this article, Kim et al describe the role of a ginsenoside Rg3-regulated lncRNA (ATXN8OS) on regulating cell growth and apoptosis. Based on previously published methylation array data, the authors identify ATXN8OS as a lncRNA that is repressed in expression by Rg3 through hypermethylation of its promoter in breast cancer cells (MCF-7). siRNA mediated knock-down of ATXN8OS led to reduced cell growth and increased apoptosis in MCF-7 cells. miR-424-5p, a miRNA predicted to target ATXN8OS, was shown to have opposite effects on cell growth and apoptosis, is controlled by Rg3 and ATXN8OS, and appears to regulate ATXN8OS abundance itself. Several potential miR-424-5p target genes, which also show promoter hypermethylation upon Rg3 treatment, appear to be regulated by ATXN8OS as well as miR-424-5p. Based on these findings, the authors suggest a role for ATXN8OS as a sponge for miR-424-5p and hypothesize that Rg3 might suppress MCF-7 cell proliferation based on ATXN8OS/miR-424-5p mediated regulation of oncogenic target genes.

In general, the topic of this study is of high interest for the field. However, several of the key experiments shown here lack crucial controls. Furthermore, some of the hypotheses and conclusions drawn in this article require additional experimental data for verification.

Briefly my comments are:

1) One of my main concerns about this study is based on the validity of the last sentence in the conclusion section (line 313-314), stating that Rg3 might suppress MCF-7 proliferation by modulating the ATXN8OS/miR-424-5p/target gene axis. Based on this and published data summarized in the introduction, Rg3 treatment alone should lead to activation of apoptosis and/or suppression of cell cycle progression (lines 34-38). Why then did Rg3 treatment of MCF-7 cells shown in the results part of this article not yield any growth defects, at least in figures 1c (control siRNA treated) and 3c (control inhibitor treated)? Treatment with control mimic however (Fig. 3a), seems to result in a slight growth defect when comparing not Ng3-treated cells vs. cells treated with 20 or 50 μM Rg3. These inconsistencies between previously reported data and data shown in this article should be explained.

2) siRNA-mediated knockdown of ATXN8OS was used to show functionality of the lncRNA throughout the article. Based on the data shown and the information given in the methods section, only one siRNA targeting ATXN8OS was used for these experiments. In order to limit the significant potential for off-target phenotypic effects, it is state of the art to use multiple different siRNAs targeting an RNA of interest. Therefore, it is crucial to verify specificity of the phenotypes shown in this study with additional siRNAs directed against ATXN8OS. Alternatively, rescue of ATXN8OS depletion by overexpression of the lncRNA (if the exact sequence /isoform is known) in the knockdown setting could also be used to verify specificity of the phenotypes.

3) The authors make several statements about a role for ATXN8OS in promoting proliferation. However, the assay used for testing proliferation is a cell viability analysis and not suitable for distinguishing increased cell proliferation from changes of cell numbers based on cell death. Therefore, proliferation assays (such as cell cycle FACS analysis) need to be performed to verify these hypotheses. This would also clarify, whether some of the changes in cell numbers (such as in Fig. 1c for ATXN8OS knockdown) could be due to increased apoptosis (such as shown in Fig. 1d).

4) Based on the data shown in Fig. 3, the authors suggest that both ATXN8OS and miR-424-5p sponge each other. However, it is not clear how knockdown of the lncRNA should result in a highly increased abundance of the miRNA as shown in Fig. 2c if the lncRNA merely acts as sponge. The overall amount of the miRNA that was titrated away should not change, unless it is degraded upon binding to the lncRNA. Clearly, additional experimental data is needed to verify sponge function of ATXN8OS and direct targeting by miR-424-5p.

5) In lines 210-212, the authors state that cell growth was suppressed by the miR-424-5p mimic alone, and was further reduced by Rg3 treatment (Fig. 3a). While miR424-5p mediated suppression of growth is clearly shown in the growth assay in Fig. 3a, it is not apparent that Rg3 would significantly further decrease growth in miR-423-5p mimic-treated cells. This statement should be re-worded or clarified.

6) In order to identify potential targets for ATXN8OS/miR-424-5p, the authors applied 2 filters: 1) miR-target gene search and 2) genes hypermethylated by Rg3 treatment. The choice for the second filter should be explained further: Why did the authors expect the ATXN8OS target genes to also be directly regulated by Rg3-mediated changes in promoter methylation? If this is actually the case, how relevant is the “ATXN8OS / miR-424-5p axis” for Rg3-mediated regulation of target genes such as EYA1 and DACH1?

Minor points:

A) Supplementary figure 1 shows knockdown tests for the siRNA targeting ATXN8OS using two concentrations (20 and 40 nM). Which concentration was used for the experiments in the main figures?

B) Figures 1d as well as 3b+d show FACS results including selected gates. These images should be labeled more clearly.

C) Line 235-236: “…showed hypermethylation in the array data, suggesting that they were downregulated by Rg3…” seems not to correspond to statement in line 232: …genes that were significantly demethylated by Rg3…” as well as the statement in lines 312-313: “…oncogenes such as Eya1, DACH1 and CHRM3, which could be activated by Rg3 treatment…”. These statements need to be re-worded to have corresponding meaning.

D) Line 233 “GSE99505” instead of “GEO99505”

Author Response

(The authors gave the same response as above.)

Reviewer 3 Report

Sun et. al. studied the anti-tumor activity of Ginsenoside Rg3  in the human and ER-positive MCF7 breast cancer cell line.  The authors demonstrate that Rg3 exerts an anti-proliferative action by down regulation of the lncRNA, ATXN8OS, via hyper-methylation of the corresponding promoter.  As a consequence, the potential ATXN8OS target microRNA, miR-424-5p, is up-regulated and acts as a tumor suppressor.  The finding that Rg3 regulates the growth and apoptosis of breast cancer cells through transcriptional regulation of this lncRNA and the miR-424-5p target is novel.  However, this might be just one of the many mechanisms underlying the anti-tumor activity of ginsenosides.  Overall, the experiments are well organized and the results are solid.  However, the work is not suitable for publication in the present form.  In particular,

  1. All the work has been performed only in one ER-positive breast cancer cell line, which is an obvious limitation of the entire study. In addition, the mechanism responsible for the tumor suppressive action of Rg3 is unlikely to be simply due to a transcriptional effect on ATXN8OS and miR-424-5p.  Indeed the chemical structure of Rg3 is similar to that of estrogens and it cannot be excluded that Rg3 exerts a direct anti-estrogenic or pro-estrogenic action, which has already been reported (Tian Mei, et. al. Chinese Journal of Natural Medicines 2020, 18:  526-535). Thus, the observation needs to be confirmed in at least another triple negative (ER-negative) breast cancer cell line, such as MDA-MB231.
  2. Although statistically significant, the anti-proliferative and apoptotic effects of siATXN8OS are very limited (15.9% and 5% inhibition, respectively; see Fig.1C-D).  In contrast, the effects of miR-424-5p mimic on cell proliferation and cell death are much more significant (Fig.3A-B).  This discrepancy might be explained by the fact that Rg3 is endowed with estrogenic properties which stimulate cell growth and partially counteract the anti-proliferative activity of  Rg3.    Therefore, I think that the experiments performed in other ER-negative cell lines is essential to strengthen the results presented in this work. 

Author Response

(The authors gave the same response as above.)

Round 2

Reviewer 2 Report

In the revised version of the manuscript by Kim et al., the authors respond to my concerns and present a significant amount of additional experimentation since the last revision. Key concerns however, were not sufficiently resolved with the newly added data:

A) In response to my concern about potential off-target effects of the only siRNA targeting ATXN8OS used in the previous version of the article, the authors synthesized a second siRNA. However, merely its effect on apoptosis was tested, and it unfortunately showed a different response in early apoptosis than siATXN8OS #1 (Fig. 1D vs. Fig. S3). It is not clear why the authors did not at least include the second siRNA for the growth rate (Fig. 1C) or cell cycle FACS analyses (Fig. 1E) – all of which were (re-)done in response to my previous suggestions.

B) In point 5 of my previous comments, I commented on the lack of an apparent effect of Rg3 treatment on cell proliferation in Fig. 3A. This issue is not resolved as it still contradicts the statement made in the text (now lines 229-231 in the new version), which states that Rg3 further decreased cell growth in a dose dependent manner, in addition to the effect mediated by the miR-424-5p-mimic. This issue is still not resolved and the answer by Kim et al. only referred to Fig. 3c, which was redone with new a cell batch and fresh Rg3 reagent. Why was the experiment shown in Fig. 3a not repeated in a similar manner to address the concern? The figure shown here still shows no further decrease of growth in a Rg3 dose dependent manner after treatment with the miR-mimic.

I appreciate the possibility that old batches of cells and/or Rg3 reagent might have caused the lack of suppressed growth mentioned in point 1 of my previous comments. If this turned out to be the case however, all experiments affected by this problem should have been repeated.

C) With respect to the question about filters used to identify potential targets for ATXN8OS/miR-424-5p, Kim et al argued that promoter hypermethylation data was used as a filter because they searched for target genes downregulated by Rg3 and because the data was already at hand. I agree that hypermethylation of promoter sites is a way of suppressing gene regulation. However, as stated in the abstract (lines 27-30) and elsewhere, the authors hypothesize that these target genes are regulated by Rg3-mediated downregulation of ATXN8OS, which in turn acts as a sponge for miR-424-5p, thus leading to downregulation of its target genes. Therefore, the mode of suppression of genes such as DACH1 appears to be post-transcriptional – mediated by a miRNA. This is in contrast to gene regulatory control of transcription based on promoter methylation. Therefore, choosing promoter methylation as a filter does not fit at all to the hypothesis proposed by the authors based on the data shown. For this reason, the selection of the second filter should have been sufficiently explained to increase confidence in the proper selection of target genes as well as the hypothesized mode of action.

Reviewer 3 Report

The authors revised the manuscript appropriately according to the comments raised by the reviewer.  Only one problem is that the newly cited paper (26, on line 167) is not correct.  Thus, the reference has to be checked carefully and corrected properly.